# Potential of Brewer’s Spent Grain as a Potential Replacement of Wood in pMDI, UF or MUF Bonded Particleboard

**DOI:** 10.3390/polym13030319

**Published:** 2021-01-20

**Authors:** Marius Cătălin Barbu, Zeno Montecuccoli, Jakob Förg, Ulrike Barbeck, Petr Klímek, Alexander Petutschnigg, Eugenia Mariana Tudor

**Affiliations:** 1Forest Products Technology and Timber Construction Department, Salzburg University of Applied Sciences, Markt 136a, 5431 Kuchl, Austria; cmbarbu@unitbv.ro (M.C.B.); z.montecuccoli@me.com (Z.M.); info@foerg-verpachtung.de (J.F.); ulrikebarbeck@gmx.at (U.B.); alexander.petutschnigg@fh-salzburg.ac.at (A.P.); 2Faculty of Furniture Design and Wood Engineering, Transilvania University of Brasov, B-dul. Eroilor nr. 29, 500036 Brașov, Romania; 3TESCAN a.s., Libušina tř. 1, 62300 Brno, Czech Republic; petr.klimek@tescan.com

**Keywords:** brewer’s spent grain, particleboard, pMDI, UF, MUF, lignocellulosic waste

## Abstract

Brewer’s spent grain (BSG) is the richest by-product (85%) of the beer-brewing industry, that can be upcycled in a plentiful of applications, from animal feed, bioethanol production or for removal of heavy metals from wastewater. The aim of this research is to investigate the mechanical, physical and structural properties of particleboard manufactured with a mixture of wood particles and BSG gradually added/replacement in 10%, 30% and 50%, glued with polymeric diisocyanate (pMDI), urea-formaldehyde (UF) and melamine urea-formaldehyde (MUF) adhesives. The density, internal bond, modulus of rupture, modulus of elasticity, screw withdrawal resistance, thickness swelling and water absorption were tested. Furthermore, scanning electron microscopy anaylsis was carried out to analyze the structure of the panels after the internal bond test. Overall, it was shown that the adding of BSG decreases the mechanical performance of particleboard, due to reduction of the bonding between wood and BSG particles. This decrease has been associated with the structural differences proven by SEM inspection. Interaction of particles with the adhesive is different for boards containing BSG compared to those made from wood. Nevertheless, decrease in the mechanical properties was not critical for particleboards produced with 10% BSG which could be potentially classified as a P2 type, this means application in non-load-bearing panel for interior use in dry conditions, with high dimensional stability and stiffness.

## 1. Introduction

The richest renewable resource worldwide is the lignocellulosic biomass [1]. It includes wood residues from tree pruning, bark, wood shavings [2], wood processing residues [3], but also food farming residues and agricultural and food waste [4].

The waste of a company is the raw material of another [5]. Beer is the fifth most consumed beverage in the world, alongside to tea, carbonates, milk and coffee [6]. In 2018, the annual world estimated production was 1.94 billion hectoliters beer [7]. In the manufacture of beer, are generated different residues and by-products. They include spent grains, spent hops and surplus yeast [8]. Brewer’s spent grain (BSG) is the main residue from breweries—with a percentage of 85% of the total by-products [9] and its share is 20 kg per 100 L of manufactured beer [9,10,11]. One drawback of the upcycling of BSG is limited especially in developing countries. New possibilities to rich out to this residue would be economically valuable [5]. It should be considered also that wet brewer’s grains are decaying quick, are voluminous and contain large amounts of water and the transport costs should be also not neglected [12]. For this reason, their distribution is limited to a radius of 150–350 km around the brewery. Dehydration or freezing facilitate the distribution beyond their area of production [13].

This lignocellulosic material is rich in oligo- and polysaccharides and polyphenols [5], with low-cost and large availability that opens new opportunities for the use of this by-product [5] and [14]. Due to the high amount and constant availability of BSG close to high populated areas many alternative forms of utilization have emerged. BSG has been used on a large scale as animal feed [5,15] for the production of bioethanol [1,10,16]. BSG may be utilized for disposable trays [12], construction bricks, metal adsorption (removal of heavy metals from wastewater) and immobilization or as a growth medium for microorganisms and enzyme [10]. There are also food applications of BSG in the manufacture of frankfurters [17], bakery products as bread, cookies, cakes and fruit beverages [18,19,20] and [9].

The manufacture of particleboard from different agricultural and agro-forest residues from walnut, almond, and pine nut shells was studied by [21,22] wheat [23], rice [24] and rapeseed [25,26], rape straw [27], straw or sunflower stalks [28]. The modification of the core layer of particleboard with hemp shives was analyzed by [29], the effect of adding sugarcane stalks and bamboo culms in PB was studied by [30] and [31] manufactured PB with sunflower, cup plant and topinambour stalks. The suitability of BSG for the manufacture of PB bonded with urea-formaldehyde resin was investigated by [14], when wood particles were replaced gradually at percentages of 10, 20, 30 and 50 with BSG. This present study is a follow-up of the research of [14] and it aims to analyze the influence of the bonding of BSG with polymeric diisocyanate (pMDI), urea-formaldehyde (UF) and melamine urea-formaldehyde (MUF) adhesives on the properties of PB with 10, 30 and 50% BSG wood particles.

## 2. Materials and Methods

The brewer’s spent grain was provided by Flötzinger Bräu (Rosenheim, Germany), Egger Co. (Unterradlberg, Austria) and Stiegel (Salzburg, Austria) breweries. The initial moisture content (m.c.) of the BSG was between 250% and 300%. The initial batch of BSG was dried at 60 °C with a Brunner-Hildebrand High VAC-S, HV-S1 (Hannover, Germany) kiln dryer for 24 h, until the target moisture content of 3% was reached. Because BSG is quite a wet material, it is expensive and difficult to be transported and could decay quick due to high percentage of polysaccharides (17% cellulose and 28% non-cellulosic sugars [9]).

The BSG was frozen in small packages (about 1.5 kg) to avoid bacterial growth which would compromise material quality. Subsequently the block of frozen BSG was chopped into smaller parts that were spread on aluminium pans. The pans were placed in a Brunner-Hildebrand High VAC-S, HV-S1 (Hannover, Germany) kiln dryer for 20 h, to reach the m.c. of 3%. After drying, BSG were stored in small bags for later use. In the case of industrial scale production, a drain extruder of screw press dewatering equipment can be employed.

BSG and spruce wood (*Picea abies*) particles were mixed with 4% pMDI type ONGRONAT WO 2750 from BorsodChem (Kazincbarcika, Hungary), 13% MUF type PRIMERE 10H119 from Metadynea (Krems, Austria) and 14% UF type Preferé 10F102 from Metadynea (Krems, Austria).

For the gluing with pMDI, the particleboards were pressed under special air exhaustion conditions in the reserch center (TechCenter) of Egger Co. in Unterradlberg, Austria. Due to the company’s guidelines, the equipment type cannot be revealed.

The 400 mm × 400 mm particleboards, with a thickness of 15 mm and densities ranging from 550 to 850 kg/m^3^ were pressed in four different groups (0% BSG, 10% BSG, 30% BSG and 50% BSG) with three replications for each board (Table 1, Figure 1).

The selected grain size for all BSG types was 2–4 mm and >4 mm. For MUF and UF 2% ammonium sulphate hardener was added.

After pressing, the particleboards were cooled, the sides and loose material was cut off, then were conditioned at 20 °C and 65% relative air humidity for one week before the testing specimens were cut [32].

The density of the panels was calculated according to [33]. Thickness swelling (TS) and water uptake (WA) after 24 h water immersion were determined according to [34]. The test samples with 50 mm × 50 mm were weighed. The thicknesses and weight were measured with 0.1 mm and 0.01 g accuracy level. The specimens were immersed in de-ionized water (Ph = 7 ± 1), at 20 °C for 24 h. During this time the samples had neither contact to each other nor to the walls of the water container. Afterwards the test specimens were removed and rinsed to eliminate excessive water. Each sample was reweighed and the sample’s size was taken from the same location previous to immersion in water.

The mechanical tests were carried out with the universal testing machine Zwick/Roell Z 250 (Ulm, Germany).

The bending strength (modulus of rupture, MOR) and the modulus of elasticity (MOE) were tested according to [35] with 350 mm × 50 mm × 15 mm samples. The modulus of elasticity and the bending strength were measured by a force which occurs in the middle of the sample.

To determine the internal bond (IB) of the test samples (50 × 50 × 15 mm), the transverse tensile strength was tested according to [36]. The testing specimens were fixed in between two testing plates with hotmelt-glue. Results that exhibited any form of glue line failure were rejected.

The screw-withdrawal (SW) resistance was made according to [37]. For this test, a wood screw (4.2 × 38 mm) was drilled in the surface of 75 × 75 mm samples. The pull-out force was measured. A hole of 3 mm diameter was predrilled in the centre of the samples. Then the fastener was screwed in 20 mm deep. The maximum force required to withdraw the screw was recorded for each testing specimen.

The scanning electron microscopy analysis (SEM) was carried out on TESCAN MIRA (Brno, Czech Republic). The morphology of control sample (100% wood, bonded with pMDI) and sample with Wood:BSG (50:50, bonded with pMDI) content was studied. Prior the morphology evaluation, samples were ruptured in pull, simulating the IB test and fractured surface has been inspected. Morphology and interaction between BSG particles and wood particles were visually evaluated, assessing type and extend of the failure. Samples were platinum-coated in a vacuum sputter coater. Accelerating voltage was set at 3 keV and beam current was 300 pA.

## 3. Results and Discussion

Dried BSG grains had the particles size distribution as per Figure 2—sieve shaker Retsch AS 200 (Haan, Germany) with six type of meshes: 4 mm, 2 mm, 1 mm, 500 μm, 315 μm and 250 μm was used.

Two things could be observed during and after pressing. The boards with BSG get stained with dark brown spots (Figure 3). It is assumed to be caused by the BSG which came in contact with the hot press plates (220 °C) and not due to the amount of sugars (cellulose and hemicellulose) in BSG, that are less than 50% [14]. Also, the particleboards with 50% BSG smelled like bread and the BSG was clearly visible on panel surface.

The m.c., tested according to EN 322:1993 (of the samples measured after conditioning (20 °C and 65% relative air humidity) of the samples glued with pMDI was 11% and for both UF and MUF bonded samples was 9%. The m.c. of wood particles was 2.5%.

The results of the ANOVA are outlined in Table 2, where are listed the factors that statistically significantly influenced the panel properties. The statistical model for the dependent variables (IB, MOR, MOE, TS, WA and SW) was highly significant (*p* < 0.001) for all variables and the explanatory power that measures the strength of the relationship between the dependent and independent variables was high, as shown by *η*^2^ values higher than 0.75 for all investigated panel properties.

The results of the mechanical properties—internal bond (IB), modulus of rupture (MOR), modulus of elasticity (MOE) and screw withdrawal resistance on surface (SW) are presented in Table 3, with standard deviation (SD) in parentheses.

### 3.1. Internal Bond

The IB of the tested specimens, measured according to EN 319:1993 [36], is highly significantly (*p* < 0.001) and it is influenced by the density of the panel, the glue type and the amount of BSG used for the manufacture of the particleboard. The distinction between significant and meaningful results is presented through regression analysis, when the explanatory power (eta squared values) explores these characteristics. The eta squared values are highest with density (0.61) and glue type (0.68), being less influenced by the BSG amount (0.07).

The maximum mean values of IB (Figure 4) were achieved in the case of UF-bonded panels: 1.18 (SD 0.12) N/mm^2^, when 10% BSG were included in the formulation of the PB. These upmost values were achieved also due to the density from 700 to 850 kg/m^3^. The trend of IB is given by the samples with 100% wood particles that have the strongest bonding, decreasing with increased amount of BSG. The trial MUF30BSG panels (density 700 kg/m^3^) makes the exception, with a mean of 0.31 (SD 0.05) N/mm^2^ for the IB, value higher than that of all other boards glued with MUF.

### 3.2. Moduli of Eupture and Elasticity

The MOR (Figure 5) and MOE (Figure 6) of the tested specimens, measured according to EN 310:1993 [35], are highly significantly (*p* < 0.001). The eta squared values are highest with glue type (0.75) and density (0.6), and minor influenced by the BSG amount (0.11). MOR (Figure 5 left) was lowest for the panels with a density between 550 and 650 kg/m^3^ (mostly the boards glued with pMDI), ranging from 7.7 (SD 1.3) N/mm^2^ for the panels manufactured with 10% BSG to 5 (SD 0.83) N/mm^2^ for a mixture of 50% BSG and 50% wood particles. The uppermost values of MOR were recorded at higher densities of the UF-bonded panels, from 19.64 (SD 1.09) N/mm^2^ to 13.21 (SD 0.44) N/mm^2^. In the middle are the MUF-glued boards, with 10.46 (SD 1.16) N/mm^2^ for 10% BSG and 11.2 (SD 1.2) N/mm^2^ for 30% BSG (see densities in Figure 5 left).

The highest average values for MOE (Figure 5 right) were detected for the panels manufactured with UF, 4000 N/mm^2^ (SD 0.03 GPa) for the PB without BSG and 3050 N/mm^2^ (SD 0.01 GPa) for an amount of 10% BSG. Scrutinizing the panels with densities less than 650 kg/m^3^, the lowest value was detected at an amount of 50% BSG—880 N/mm^2^ (SD 0.01 GPa). The values of the MUF-bonded PB were included between 2010 N/mm^2^ (SD 0.06 GPa) (no BSG in composition) and 2700 N/mm^2^ (SD 0.02 GPa) (30% BSG).

### 3.3. Screw Withdrawal Resistance

It is accepted by experiments that certain mechanical properties, from that MOE and MOR of wood-based composites are correlated to screw withdrawal resistance, measured according to EN 320:1993 [38]. That means that the same tendencies observed in 3.1 and 3.2 are applicable also for SWR (Figure 7), which is dependent on glue type [39], with a value of eta squared of 0.62 and less influenced on density type (0.3) and BSG amount (0.22). The SWR in the surface measured at the surface of panels glued with UF and with a density up to 850 kg/m^3^ was between 76 and 143 N/mm and similar to the values obtained for the MUF-glued panels (92 to 116 N/mm). A half of these values were measured for lower density panels (<650 kg/m^3^) bonded with pMDI (30 to 63 N/mm).

### 3.4. Thickness Swelling and Water Absorption after 24 h

Thickness swelling (TS) after 24 h (Table 4), measured according to EN 317:1993 [34] of the particleboard with 10%, 30% and 50% BSG was significantly higher than that of all panels with 100% content of wood particles (*p* < 0.05) for all types of adhesives.

Small variations in the TS 24 h (Figure 8) are attributed to density (8%), BSG amount (13%) and glue type (25%). As a consequence, the panels with the highest density, up to 850 kg/m^3^, did not swelled in thickness so much as the panels with a density ranged from 650 to 750 kg/m^3^. The use of pMDI kept the TS 24 h between 17% (SD 1.71%) (samples without BSG) and 30% (50% BSG content, SD 0.8%). The TS of the UF-bonded PB is similar to the one observed for the group glued with pMDI. Slightly higher TS 24 h were measured for the PB manufactured with MUF, with a maximum of 34% (SD 2.65%).

WA after 24 h (Figure 9), measured according to EN 317:1993 (34) is less influenced by the BSG amount (18%) and in an equal measure by the density and glue amount (33%). The lowest WA was detected in the case of PB manufactured with UF (from 58 to 63%) and the highest when PB were produced with pMDI (105% at 50% amount of BSG). 

When using MUF, the percentages of WA 24 h are ranging from 80% (for PB without BSG) to 3% (at 10%BSG). 

### 3.5. Morphological Evaluation

The microscopic evaluation of the different particleboard types (Figure 10 and Figure 11) indicated potential reasons for the mechanical property differences. While structural SEM inspection of a PB made from wood showed mainly structural failures in wood (Figure 10A,B) or failures at the wood-adhesive bond interphase (Figure 10C), these were very seldom observed at the PB containing BSG particles. It was observed that BSG particles, unlike particles of wood, are frequently present in voids and pores within the particleboard structure (Figure 11A), with finer BSG particles also adhering onto wooden particles surfaces (Figure 11B) restricting proper wood particle-particle structural bonding. Furthermore, often BSG particles showed extensive porosity (Figure 11C), causing local overconsumption of adhesive which filled these pores instead of adhering to the particles surface, with the consequence of restricting proper bonding contact among the particles. We assume that aforementioned observations are connected to the actual negative effect of the BSG content on the internal bond strength of the panels.

## 4. Conclusions

The results of this study have revealed that an amount of 10% BSG for the manufacture of PB bonded with pMDI, UF and MUF results in panels with adequate mechanical and physical properties. IB, MOR and MOE met the requirements for P2, non-load-bearing panel for interior use in dry conditions with high stiffness, [40] with 0.35 N/mm^2^, 11 N/mm^2^ and 1600 N/mm^2^ respectively for mostly all types of panel sets presented in this study.

An increased volume of BSG (>20%) is reducing visible the properties of the panels. The use of two types of raw materials in the matrix of the PB is influenced by the chemical composition and gluability of its components [14]. It was interesting to find out that the bonding with pMDI did not have an increased influence on better performances regarding dimensional stability or mechanical properties. The UF-bonded panels showed the most bearable results also due to increased density (up to 850 kg/m^3^).

The SEM analysis proven that actual structural difference of panels containing BSG and those made from particles cause reduction in particle-particle bonding and it seems to be an overall reason for the reduced mechanical performance

To improve the efficiency of renewable resource use and the upcycling of a waste product as BSG, higher amounts of BSG should be further studied in combination with innovative glues. Sustainable adhesives as casein [41] or tannin-based [42] can be also introduced, to improve TS and WA with the disadvantage of long pressing times (casein) or limited shelf life (tannin) hence extended manufacturing costs [41] and limited workability [43].

## Figures and Tables

**Figure 1 polymers-13-00319-f001:**
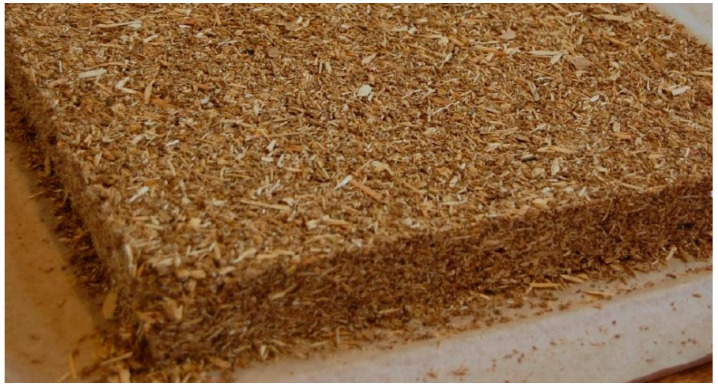
Particleboard matt single layered with 50% BSG after forming before hot pressing.

**Figure 2 polymers-13-00319-f002:**
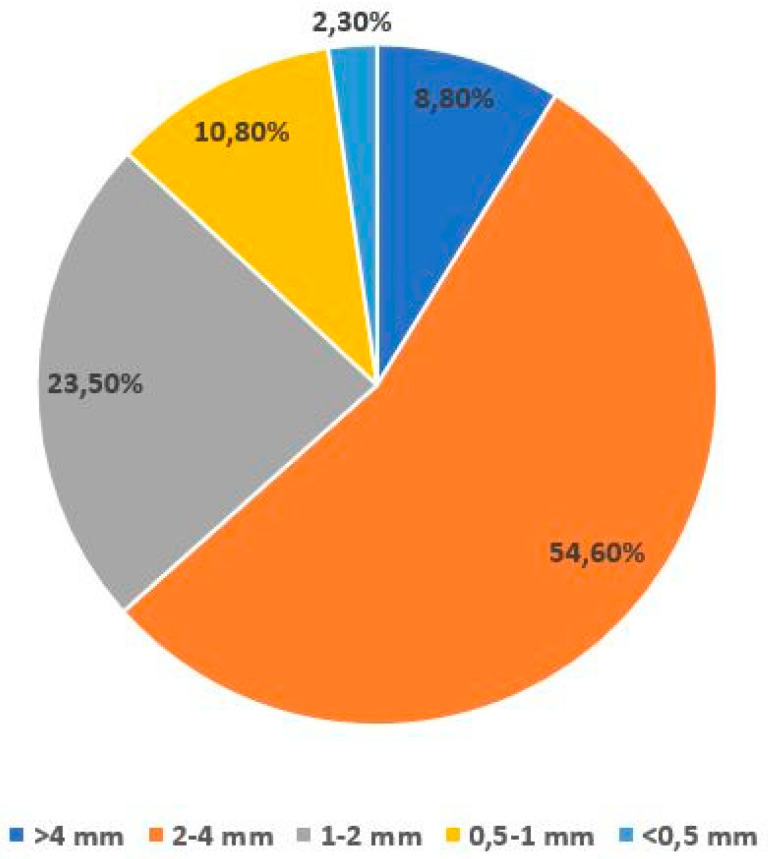
Particle size distribution of dried BSG.

**Figure 3 polymers-13-00319-f003:**
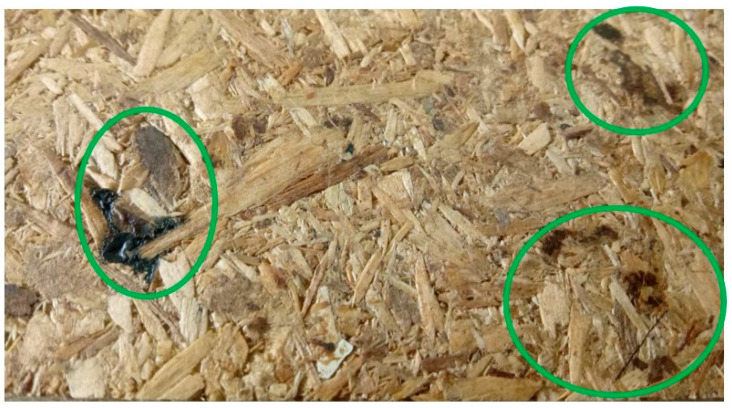
Dark brown spots on the surface of the panels with 50% BSG and 50% wood particles bonded with pMDI.

**Figure 4 polymers-13-00319-f004:**
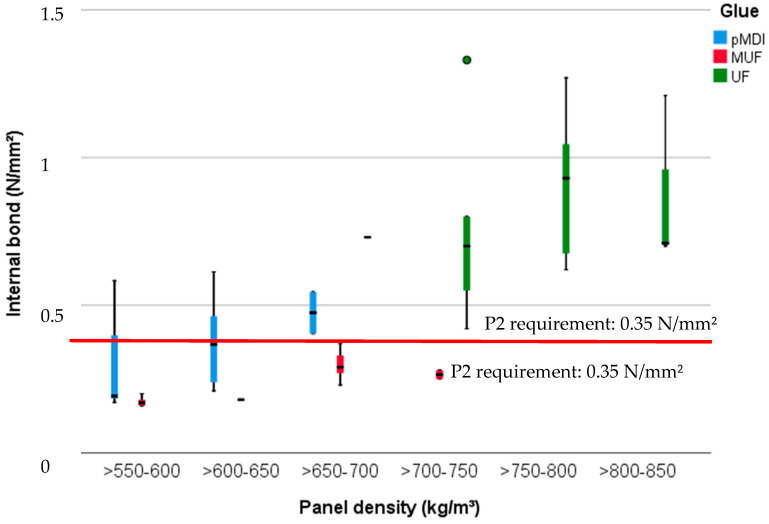
Internal bond of the 15 mm PB samples with densities between 500 and 850 kg/m^3^ and glued with pMDI, UF and MUF.

**Figure 5 polymers-13-00319-f005:**
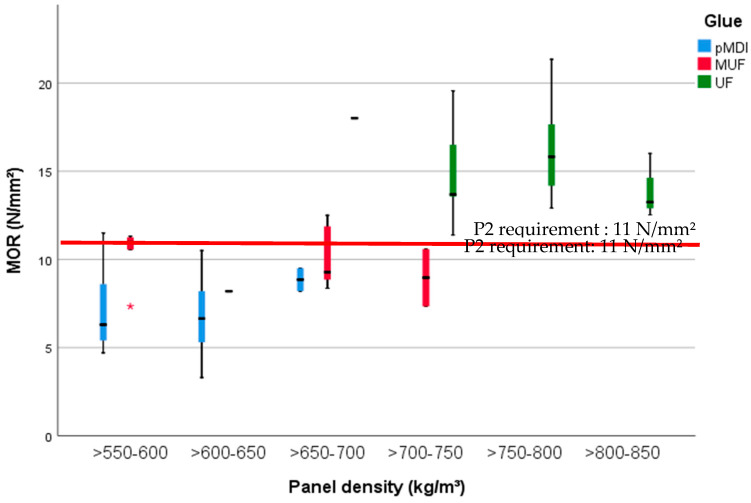
MOR of the 15 mm PB samples with densities between 500 and 850 kg/m^3^ and glued with pMDI, UF and MUF.

**Figure 6 polymers-13-00319-f006:**
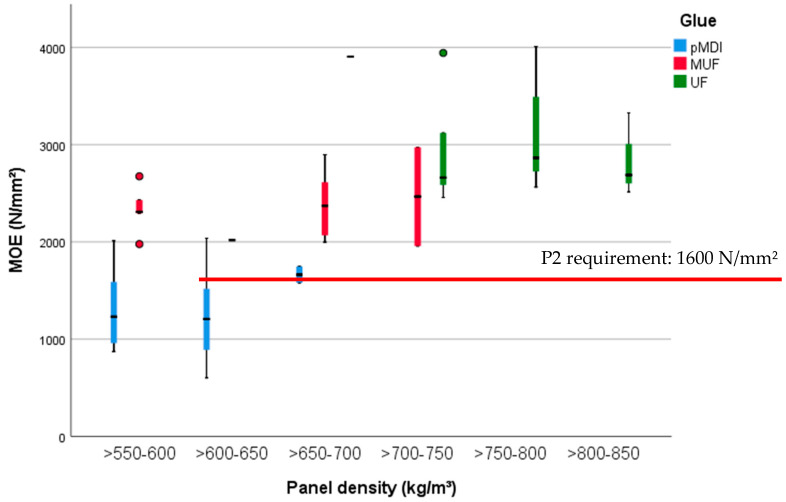
MOE of the 15 mm PB samples with densities between 500 and 850 kg/m^3^ and glued with pMDI, UF and MUF.

**Figure 7 polymers-13-00319-f007:**
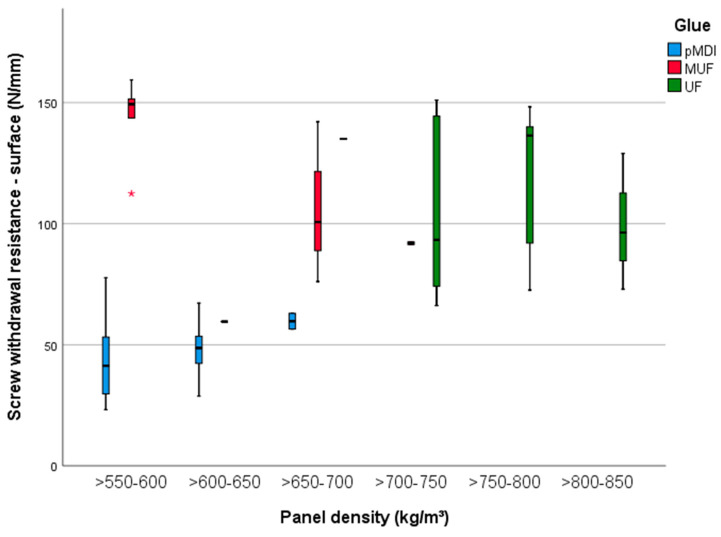
Screw withdrawal resistance measured on the surface of the 15 mm PB samples with densities between 500 and 850 kg/m^3^ and glued with pMDI, UF and MUF.

**Figure 8 polymers-13-00319-f008:**
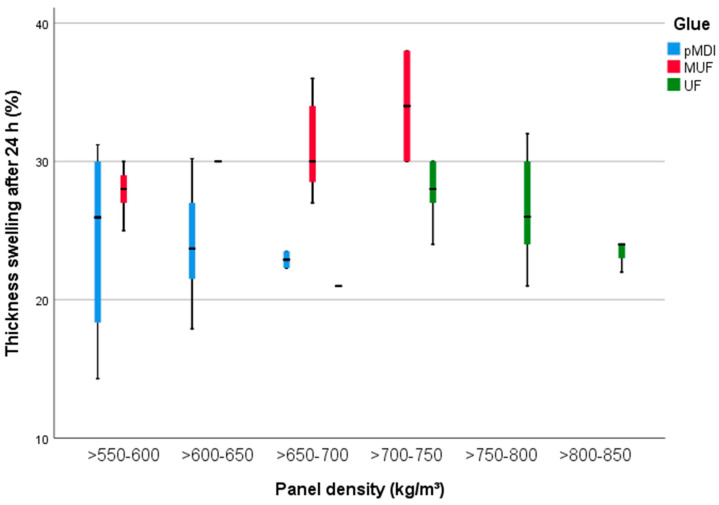
Thickness swelling after 24 h of the 15 mm PB with densities between 500 and 850 kg/m^3^ and glued with pMDI, UF and MUF.

**Figure 9 polymers-13-00319-f009:**
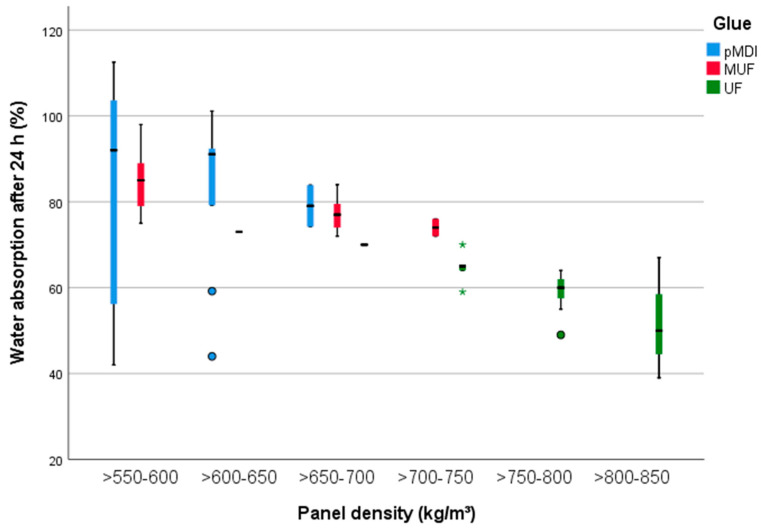
Water absorption after 24 h of the 15 mm PB samples with densities between 500 and 850 kg/m^3^ and glued with pMDI, UF and MUF.

**Figure 10 polymers-13-00319-f010:**
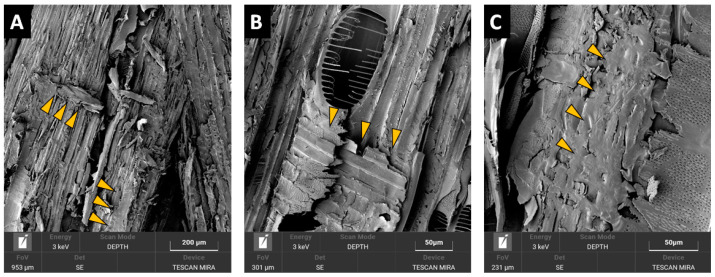
SEM images of the sample from PB board made from wood particles bonded with 4% pMDI (arrows indicating different failures).

**Figure 11 polymers-13-00319-f011:**
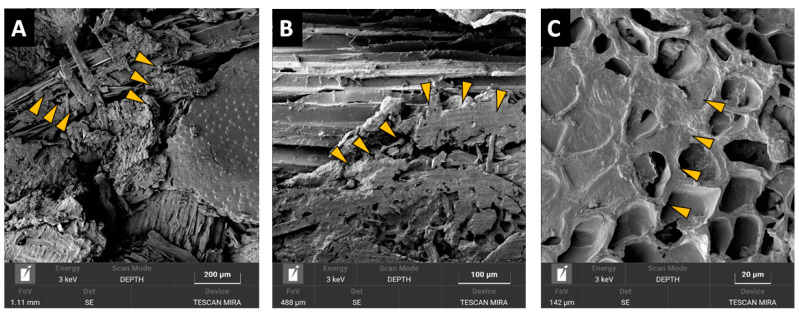
SEM images of the sample from PB board with 50% BSG bonded with 4% pMDI (arrows indicating different failures and adhesive presence in pores).

**Table 1 polymers-13-00319-t001:** Experimental design for the PB with BSG bonded with pMDI, MUF and UF.

Board	Density (kg/m^3^)	Glue Type	Glue Amount (%)	Moisture Content (%)	Press Temperature (°C)	Press Time (s)
pMDI0BSG	600	pMDI	4	11	220	130
pMD10BSG	630	pMDI	4	11	220	130
pMD30BSG	600	pMDI	4	11	220	130
pMD50BSG	600	pMDI	4	11	220	130
UF0BSG	750	UF	14	9	180	450
UF10BSG	750	UF	14	9	180	450
UF30BSG	750	UF	14	9	180	450
UF50BSG	790	UF	14	9	180	450
MUF0BSG	670	MUF	13	9	180	450
MUF10BSG	570	MUF	13	9	180	450
MUF30BSG	690	MUF	13	9	180	450

**Table 2 polymers-13-00319-t002:** Results of the ANOVA with *p*-values and *η*^2^-values for the explanatory variables.

	IB		MOR		MOE		TS		WA		SW	
	*p*	*η* ^2^	*p*	*η* ^2^	*p*	*η* ^2^	*p*	*η* ^2^	*p*	*η* ^2^	*p*	*η* ^2^
Model	0.000	0.75	0.000	0.82	0.000	0.78	0.000	0.84	0.000	0.75	0.000	0.88
Density type	0.000	0.61	0.000	0.59	0.000	0.60	0.147	0.08	0.000	0.33	0.000	0.31
Glue	0.000	0.68	0.000	0.75	0.000	0.75	0.127	0.25	0.000	0.33	0.001	0.62
BSG content	0.106	0.07	0.020	0.11	0.013	0.11	0.022	0.13	0.001	0.18	0.002	0.22

**Table 3 polymers-13-00319-t003:** Results of the physical and mechanical properties of the 15 mm particleboards (a, b, c, d, e, f, g values with the same letter are not significantly different: ANOVA, Post-Hoc Tukey HSD, α = 0.05).

Sample	Density (kg/m^3^)	IB (N/mm^2^)	MOR (N/mm^2^)	MOE (N/mm^2^)	SW (N/mm)
pMDI0BSG	600	0.48 ^a^ (0.14)	9.67 ^a^ (1.4)	1787 ^a^ (232.2)	63 ^a^ (8.95)
pMDI10BSG	630	0.44 ^a^ (0.07)	7.7 ^b^ (1.3)	1425 ^b^ (244.4)	54 ^a^ (5.94)
pMDI30BSG	600	0.22 ^b^ (0.05)	5.82 ^c^ (1.08)	1127 ^c^ (187)	41 ^b^ (5.92)
pMDI50BSG	600	0.21 ^b^ (0.02)	5.1 ^c^ (0.83)	882 ^d^ (134)	30 ^b^ (4.05)
UF0BSG	750	0.91 ^c^ (0.14)	19.64 ^c^ (1.09)	3967 ^e^ (38.77)	143 ^c^ (5.93)
UF10BSG	750	1.18 ^c^ (0.12)	16 ^d^ (0.22)	3057 ^e^ (157.7)	139 ^c^ (6.67)
UF30BSG	750	0.65 ^d^ (0.05)	13.67 ^e^ (0.98)	2666 ^f^ (103.5)	90 ^d^ (7.91)
UF50BSG	790	0.65 ^d^ (0.12)	13.21 ^e^ (0.44)	2599 ^f^ (60.54)	76 ^e^ (10.32)
MUF0BSG	670	0.26 ^b^ (0.03)	8.17 ^b^ (0.74)	2013 ^g^ (62.28)	92 ^f^ (11.76)
MUF10BSG	570	0.17 ^b^ (0.01)	10.46 ^a^ (1.16)	2345 ^g^ (212.5)	133 ^c^ (36.9)
MUF30BSG	690	0.31 ^e^ (0.05)	11.2 ^f^ (1.2)	2693 ^f^ (219.4)	116 ^d^ (17.82)

**Table 4 polymers-13-00319-t004:** Results of the thickness swelling and water uptake of the 15 mm particleboards (a, b, c, d values with the same letter are not significantly different: ANOVA, Post-Hoc Tukey HSD, α = 0.05).

Sample	TS 24 h (%)	WA 24 h (%)
pMDI0BSG	17 ^a^ (1.71)	51 ^a^ (7.26)
pMDI10BSG	23 ^b^ (1.05)	84 ^b^ (6.45)
pMDI30BSG	26 ^b^ (1.16)	92 ^b^ (2.40)
pMDI50BSG	30 ^c^ (0.80)	105 ^c^ (4.47)
UF0BSG	26 ^b^ (2.65)	60 ^a^ (6.72)
UF10BSG	23 ^b^ (1.16)	58 ^a^ (4.87)
UF30BSG	30 ^c^ (1.63)	63 ^a^ (1.77)
UF50BSG	26 ^b^ (2.72)	59 ^a^ (11.20)
MUF0BSG	34 ^d^ (2.65)	80 ^b^ (4.62)
MUF10BSG	28 ^c^ (1.729	83 ^b^ (9.30)
MUF30BSG	29 ^c^ (1.16)	75 ^b^ (2.48)

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
