# Peer review of "Potential of Brewer’s Spent Grain as a Potential Replacement of Wood in pMDI, UF or MUF Bonded Particleboard"

_polymers, 2021, doi:10.3390/polym13030319_

Round 1

Reviewer 1 Report

please see my comments in the attached file

Author Response

Dear Reviewer,

Thank you for your effort to analyse the manuscript and providing specific and detailed advice on how to improve the paper’s quality.

The manuscript was improved following your suggestions. Please find our comments next to your remarks in the pdf-file.

In the revised manuscript the changes are highlighted with blue colour and made with track-changes function.

The authors

Reviewer 2 Report

Potential of Brewer's spent grain as a replacement of wood in 2 pMDI, UF or MUF bonded particleboard

Marius Cătălin Barbu, Zeno Montecuccoli, Jakob Förg, Ulrike Barbek, Petr Klímek, Alexander Petutschnigg, Eugenia Mariana Tudor*

The overall assessment of work.

The use of annual plants rich in cellulose is a constant stream of research on modern technologies of particleboard production for the furniture industry, buildings, and many other industries. The brewer's spent grain (BSG) is also widely used. In this work, the authors undertook the task of determining the mechanical, physical, and structural properties of particleboards made of a mixture of wood and BSG particles, glued with polymer diisocyanate (pMDI), urea-formaldehyde (UF), and melamine-urea-formaldehyde (MUF) adhesives. Undoubtedly, the direction of research, their cognitive and utilitarian nature constitute a valuable extension of knowledge about new composites made of annual plants.

Specific remarks:

  • BSG samples were carefully prepared in accordance with the rules for bioactive materials. During the plates' pressing, the authors noticed that the BSG plates were stained with dark brown stains. Are the described facts resulting from excessive temperature, too many non-cellulosic sugars, or too long pressing time?
  • I propose to describe figures 4-8 more clearly. For example, the influence of the BSG content is not visible from them. We can only see the influence of density, but it is not known for which BSG shares. Besides, individual points (dashes) labeled with numbers appear in the figures. What does it mean?
  • Morphological studies are very interesting and explain most of the reasons why worse mechanical properties characterize samples with a high content of BSG.
  • The conclusions are correct and follow straight from the research.

Author Response

Dear Reviewer,

Thank you for your effort to read the manuscript and providing specific advice on how to improve the paper’s quality.

The following details were improved following your suggestions. Please find our comments next to your remarks.

Specific remarks:

  1. BSG samples were carefully prepared in accordance with the rules for bioactive materials. During the plates' pressing, the authors noticed that the BSG plates were stained with dark brown stains. Are the described facts resulting from excessive temperature, too many non-cellulosic sugars, or too long pressing time?

R1: Thank you for this issue, we have added the information that these stains were caused by the temperature of 220°C and not due to the sugars in BSG (less than 50%, according to Klimek et al (2017)

  1. I propose to describe figures 4-8 more clearly. For example, the influence of the BSG content is not visible from them. We can only see the influence of density, but it is not known for which BSG shares. Besides, individual points (dashes) labelled with numbers appear in the figures. What does it mea

R2: The figures 4-8 describe the values of IB, MOR, MOE, SW, TS and WS depending on density (from 550 to 850 kg/m³. The influence of BSG content is presented in the Tables 3 and 4. The boxplots diagrams were special depicted like this, because the authors did not want to present redundant information. The values of individual points were removed, in order to not create confusion about the values. These individual points are the outliers.

  1. Morphological studies are very interesting and explain most of the reasons why worse mechanical properties characterize samples with a high content of BSG.

Thank you for your remark.

The conclusions are correct and follow straight from the research.

Thank you for the assessment of our research.

The authors

Round 2

Reviewer 1 Report

The comments have been successfully addressed and therefore I am happy to suggest acceptance of the paper in its revised form